# Laryngopharyngeal Reflux: A State-of-the-Art Algorithm Management for Primary Care Physicians

**DOI:** 10.3390/jcm9113618

**Published:** 2020-11-10

**Authors:** Jerome R. Lechien, Sven Saussez, Vinciane Muls, Maria R. Barillari, Carlos M. Chiesa-Estomba, Stéphane Hans, Petros D. Karkos

**Affiliations:** 1Department of Human Anatomy and Experimental Oncology, Mons School of Medicine, UMONS Research Institute for Health Sciences and Technology, University of Mons (UMons), B7000 Mons, Belgium; sven.saussez@umons.ac.Be; 2Department of Otolaryngology-Head & Neck Surgery, Foch Hospital, School of Medicine, UFR Simone Veil, Université Versailles Saint-Quentin-en-Yvelines (Paris Saclay University), 92150 Paris, France; prhans.foch@gmail.com; 3Department of Otolaryngology-Head & Neck Surgery, Ambroise Paré Hospital, APHP, Paris Saclay University, 92150 Paris, France; 4Department of Otolaryngology-Head & Neck Surgery, CHU Saint-Pierre, Faculty of Medicine, University Libre de Bruxelles, 1000 Brussels, Belgium; 5Department of Otolaryngology-Head & Neck Surgery, CHU Ambroise Paré, 92150 Paris, France; 6Division of Gastroenterology and Endoscopy, CHU Saint-Pierre, Faculty of Medicine, University Libre de Bruxelles, 1000 Brussels, Belgium; Vinciane_muls@stpierre-bru.be; 7Division of Phoniatrics and Audiology, Department of Mental and Physical Health and Preventive Medicine, University of Naples SUN, 34103 Naples, Italy; Mariarosariabarillari@unicampania.it; 8Department of Otorhinolaryngology-Head & Neck Surgery, Hospital Universitario Donostia, 00685 San Sebastian, Spain; chiesaestomba86@gmail.com; 9Department of Otorhinolaryngology and Head and Neck Surgery, AHEPA University Hospital, Thessaloniki Medical School, 54621 Thessaloniki, Greece; pkarkos@aol.com

**Keywords:** reflux, laryngopharyngeal, gastroesophageal, primary care, physician, management, general, treatment, diagnosis

## Abstract

Laryngopharyngeal reflux (LPR) is a common disease in the general population with acute or chronic symptoms. LPR is often misdiagnosed in primary care because of the lack of typical gastroesophageal reflux disease (GERD) symptoms and findings on endoscopy. Depending on the physician’s specialty and experience, LPR may be over- or under-diagnosed. Management of LPR is potentially entirely feasible in primary care as long as General Practitioners (GPs) are aware of certain “red flags” that will prompt referral to a Gastroenterologist or an Otolaryngologist. The use of patient-reported outcome questionnaires and the consideration of some easy ways to diagnose LPR without special instrumentation oropharyngeal findings may help the GP to diagnose and often manage LPR. In this review, we provide a practical algorithm for LPR management for GPs and other specialists that cannot perform fiberoptic examination. In this algorithm, physicians have to exclude some confounding conditions such as allergy or other causes of pharyngolaryngitis and “red flags”. They may prescribe an empirical treatment based on diet and behavioral changes with or without medication, depending on the symptom severity. Proton pump inhibitors and alginates remain a popular choice in order to protect the upper aerodigestive tract mucosa from acid, weakly acid and alkaline pharyngeal reflux events.

## 1. Definition

In 2002, the American Academy of Otolaryngology Head and Neck Surgery defined Laryngopharyngeal Reflux (LPR) as the backflow of stomach contents into the laryngopharynx [1]. This definition of LPR has recently been considered incomplete because the irritation from LPR due to pepsin, bile salts and other gastroduodenal proteins does not involve only laryngopharyngeal mucosa but extends to all upper aerodigestive tract mucosa [2]. LPR is often involved in the development of many laryngeal [3], rhinological [4] and otological [5,6] conditions. Currently, LPR may be defined as an inflammatory condition of the upper aerodigestive tract tissues related to the direct and indirect effect of gastric or duodenal content reflux, inducing morphological changes in the upper aerodigestive tract [2]. In practice, we may consider two types of LPR according to the evolution of complaints over therapeutic or non-therapeutic time: acute and chronic LPR. Acute LPR may consist of the sporadic development of LPR, which is well-treated with an adequate treatment. Patients with acute LPR have no chronic course of the symptoms. Chronic LPR may concern patients with chronic course of the LPR-symptoms with a lack of or a poor therapeutic response or frequent recurrences of symptoms over time (>2 episodes yearly) requiring repeated therapeutic trials. In both definitions, LPR may be diagnosed with objective testing or empirical treatment.

This paper aims to overview the current literature about LPR epidemiology, diagnosis and treatment. Based on the recent literature findings, we aim to provide practical findings and clinical algorithm for non-otolaryngologist and primary care physicians to manage LPR.

## 2. Epidemiology

### 2.1. Prevalence and Incidence

For LPR, hypopharyngeal-esophageal intraluminal impedance-pH monitoring (HEMII-pH) is considered the diagnostic gold standard. To date, there is no study evaluating the prevalence or incidence of LPR with HEMII-pH in the general population or in an Ear, Nose and Throat (ENT) outpatient clinic.

In 1991, Jamie Koufman estimated the LPR incidence at 10% of a general ENT outpatient clinic. [7]. Koufman found that 30% of patients had documented an acid pharyngeal reflux event based on dual-probe pH monitoring. At the same time, Gaynor evaluated that 1% of patients who visited primary care physician had symptoms suggestive of LPR, but no testing was performed to confirm the diagnosis [8]. The prevalence of LPR-related symptoms in the general population was evaluated in other studies through patient-reported outcome questionnaires and ranged from 5 to 30% of cases [9,10,11]. Based on geographical, diet and lifestyle habits variations, it is estimated that LPR-symptoms could be found in 5 to 30% of individuals [2].

### 2.2. Is LPR Over- or Under-Diagnosed?

LPR is often considered as over- or under-diagnosed. In practice, because the symptoms and findings are both nonspecific [12], the detection of LPR is still complicated. According to some reports [13,14], LPR would be over-diagnosed, especially as the cause of hoarseness. In a chart-review of 105 voice clinic patients, Thomas et al. observed that dysphonia was often mis-attributed to LPR in patients with unapparent vocal fold abnormalities [13,14]. On the contrary, some physicians believe that LPR is over-diagnosed. In a recent paper, Frazer-Kirk reminded us that LPR is a common cause of upper aerodigestive tract disorders but may be under-diagnosed due to the lack of awareness of the clinical differences between GERD and LPR [15].

In practice, the risk to over- or under-diagnose LPR depends on many factors, including experience and knowledge about LPR symptoms and findings [16], physician’s expertise [17] and naturally, the diagnostic method. It is commonly recognized that the over-evaluation of LPR signs and symptoms may be responsible for overdiagnosis [18], which strengthens the need to base diagnosis on an approach that is as objective as possible.

## 3. Pathophysiology

### 3.1. LPR is not GERD and GERD is not LPR

Many practitioners usually believe that “*if the patient has no heartburn or GERD symptoms, there is no LPR*”. Several studies demonstrated that LPR is not an extension of the lower esophageal refluxate into the upper aerodigestive tract. The Montreal criteria defined GERD as a condition that develops when the reflux of stomach contents causes troublesome symptoms and/or complications such as esophagitis [19,20]. The diagnosis may be investigated with pH study that, according to Johnson et DeMeester, has to report a length of time >4.0% of the 24-h recording spent below pH 4.0 or a DeMeester score >14.72 [21]. In practice, patients with LPR may not have heartburn, esophagitis and they usually do not meet the GERD criteria of diagnosis at the pH study [22,23]. It is assumed that ≤50% of LPR patients have GERD [24,25], while laryngopharyngeal complaints were present in 32.8% of GERD patients [25]. LPR patients mainly have gaseous, upright and daytime reflux events, and only 5.5% of pharyngeal reflux events occurred at nighttime and were recumbent [26]. The gastrointestinal endoscopy may be normal in more than 44% of cases and may reveal esophagitis in 10 to 30% of LPR patients [26,27,28], whereas erosive esophagitis is found in almost 50% of GERD patients [29]. Barrett metaplasia is still rare in LPR patients [26,27,28]. Patients with Barrett’s metaplasia had, however, a higher rate of LPR than those with mild erosive esophagitis [30,31]. Others observed that patients with esophagitis have LPR findings in 24% of cases [32]. There is a correlation between the severity of GERD and the development of LPR [30]. There are no universally accepted criteria for LPR definition, although some authors agree with the need to have more than 1 pharyngeal acid, weakly acid or alkaline reflux episode at the HEMII-pH [2,22,23].

### 3.2. Pathogenesis and Pathophysiology (I Think This Whole Paragraph Can Be Omitted)

#### Many Grey Areas

The pathophysiology of LPR is still incompletely understood. Four main research areas remain uninvestigated. First, the vast majority of research focuses on pepsin, but other enzymes can play a key role in the development of the mucosal inflammatory reaction. Few studies support the refluxate of bile salts [33,34] without providing clear conclusion about the place of bile salts in the inflammatory process. Second, the stress and autonomic nerve dysfunction are probably involved in the development of LPR. The autonomic nerve dysregulation may lead to the increase of the opening of LES and UES, and related pharyngeal reflux events. Currently, only a few authors identified that LPR patients had autonomic nerve dysfunction, anxiety or stress [35,36]. Third, laryngopharyngeal microbiota are important for the upper aerodigestive tract’s homeostasis. As for the lower digestive tube, bacteria have a critical role in the inflammation and the mucosa regeneration through the releasing of local anti-inflammatory molecules [37]. Currently, the role of LPR and the gastroduodenal refluxate on the microbiota remains unknown. This topic was, however, studied for GERD, metaplasia and esophageal microbiota [38], providing interesting findings such as the microbiota alteration by long-term proton pump inhibitor therapy [39]. Fourth, patients with similar HEMII-pH features may not develop a similar clinical picture. The interindividual differences in the laryngopharyngeal mucosa sensitivity are probably one of the most important factors underlying these clinical differences. To date, the laryngopharyngeal hypersensitivity condition was poorly studied in LPR disease.

## 4. Clinical Picture

### Symptoms

The most prevalent symptoms associated with LPR are globus sensation, throat clearing, hoarseness, excess throat mucus or postnasal drip [2,12,40]. These symptoms, which are commonly observed in primary care medicine, are nonspecific and may be associated with active laryngopharyngeal allergy [2], rhinitis [41], chronic rhinosinusitis [42], smoking [43], alcohol abuse [44] and benign laryngopharyngeal infections [45]. In other words, it is difficult to diagnose LPR only based on symptoms [46,47]. Belafsky et al. developed in 2001 the reflux symptom index (RSI) [48]. An RSI is a nine-item patient reported outcome questionnaire assessing the severity of symptoms. An RSI>13 was identified as suggestive of LPR. The mean weaknesses of the RSI are the lack of consideration of some prevalent symptoms, such as throat pain, odynophagia, halitosis or regurgitations, and the lack of consideration of the symptom frequency [12]. For these reasons, reflux symptom score (RSS), which is a 22-item patient reported outcome questionnaire, was recently developed [49]. RSS considers the most prevalent otolaryngological, digestive and respiratory symptoms and evaluates symptom frequency, severity and the potential impact on quality of life. Patients fulfill the RSS in 1 to 2 mins, which may be considered as a longer time for physicians. For this reason, based on the most prevalent and relevant findings identified in large cohort studies using RSS [23,26,49], a short version of RSS, the RSS-12, was developed [50]. RSS-12 consists of a 12-item clinical tool assessing both frequency and severity of the most prevalent LPR-related symptoms as well as their impact on quality of life (Table 1). RSS and RSS-12 reported better discriminative properties than RSI [49,50]. An RSS-12>11 is suggestive of LPR and is a practical clinical tool that may be used in general medicine to monitor the symptom evolution throughout therapeutic course. For patients with digestive complaints, the use of RSS, which include digestive items, makes sense.

## 5. Findings

The most prevalent findings associated with LPR include posterior commissure hypertrophy, arytenoid erythema and oropharyngeal and anterior pilar erythema. As for symptoms, there may have been discrepancies between prevalence and the thoughts of physicians. In 2001, Belafsky et al. developed a reflux finding score (RFS) that rates the laryngeal findings associated with LPR [51]. RFS focuses on laryngeal findings and does not consider extra-laryngeal findings. The interrater reliability of RFS is low, especially regarding the non-specificity of signs [52], which limits the reproducibility between otolaryngologists [16]. As reported by Hicks et al., some LPR-associated findings may be found in normal individuals [52]. In order to bypass the weaknesses of RFS and the lack of a clinical instrument which takes into consideration both laryngeal and extra-laryngeal findings [53], the reflux sign assessment (RSA) was developed (Appendix A). RSA is a 16-item clinical instrument assessing LPR laryngeal and extra-laryngeal findings. RSA is probably better than RFS [54], as it identifies both oropharyngeal and oral signs frequently associated with LPR. LPR patients have a significantly higher prevalence of anterior pilar erythema, coated tongue, uvula and oropharyngeal posterior wall erythema compared with healthy individuals [54]. These signs may be easily seen by the primary care physician and may be useful for both the diagnosis and the posttreatment follow-up (Figure 1) [55]. However, the primary care physician has to keep in mind that the assessment of findings is still subjective supporting that the pre- to posttreatment evaluation needs to be performed by the same physician.

RSS-12 and the identification of these oral and oropharyngeal findings may be both used in primary care medicine for the diagnosis and the evaluation of therapeutic response. The primary care physician has to be aware of the LPR signs and symptoms in children or adults with chronic dental disorders, i.e., decays or erosion, with regards to the potential association between LPR and these common conditions [56,57].

## 6. Red Flags for the General Physician (GP)

### 6.1. When to Refer the LPR Patient to the Otolaryngologist?

As GPs are often the first line physicians, they should be able to recognize certain *red flag symptoms that prompt a specialist referral*. LPR symptoms or findings in smokers and alcohol drinkers require referral to ENT for fiberoptic laryngoscopy to exclude malignancy. Symptoms such as dysphonia, dyspnea, hemoptysis, neck nodes, weight loss and referred otalgia are crucial. History has to differentiate LPR-related ear or throat pain from ear or throat pain in a suspected context of malignancy. Moreover, all physicians have to keep in mind that patients who respect the anti-reflux diet often lose weight.

It is important to keep in mind that in patients with a history of chemo/radiation, salivary gland function and hydration of upper aerodigestive tract mucosa may be compromised. However, the development of new or unusual symptoms in patients with a history of head and neck cancers or radiation may be considered as another red flag. Some reports supported that the mucosa inflammation related to reflux may lead to dysphagia and aspirations [58,59], especially in elderly patients who suffered from presbyphagia. These patients may benefit from an ear, nose and throat consultation to identify the occurrence of aspirations and to prevent the related risk of pneumonia.

### 6.2. When to Refer the LPR Patient to a Gastroenterologist?

The majority of gastroenterologists commonly manage LPR patients who may have both gastrointestinal (GI) and LPR symptoms. Some conditions have to be considered as red flags and may require a GI examination. As LPR, severe GERD, esophagitis and Barrett metaplasia are occasionally linked, [30,31], LPR patients with heartburn or non-cardiac chest pain should undergo a GI endoscopy. The identification of this red flag is, however, more complicated in elderly patients who may have esophagitis or Barrett metaplasia without symptoms [60,61]. In this respect, chronic symptoms in patients >50 years should be evaluated by a specialist. The occurrence of recurrent regurgitations, hypersalivation, weight loss or GI bleeding are other red flags supporting the realization of GI endoscopy to exclude esophageal lesion, dysmotility, Zenker diverticulum or other dysmotility diseases. Patients without responses to an empirical treatment based on PPI and alginate and those with a family history of upper GI cancer also have to be evaluated in gastroenterology [62,63]. The main red flags that have to be addressed in patient otolaryngology or gastroenterology are summarized in Table 2.

## 7. Additional Examinations

HEMII-pH detects esophageal bolus movement by the measurement of changes of electrical resistance and may measure the pH of the refluxate from the esophagus to the pharynx. HEMII-pH is usually well tolerated and may represent a cost-effective approach [64]. The indications of HEMII-pH are not standardized. HEMII-pH is often used in non-responder patients to an empirical therapeutic trial or those with many confounding factors (allergy, chronic rhinosinusitis, etc.). The use of HEMII-pH in patients with moderate-to-severe LPR symptoms is increasingly considered as a cost-effective approach because that allows the prescription of personalized treatment considering the LPR features (acid, weakly acid or alkaline; upright/daytime versus supine/nighttime) [64]. Such treatment is associated with good outcomes, the possibility to reduce drug doses throughout the treatment and drug weaning at 3 to 9 months [55]. The pharyngeal event may be detected by oropharyngeal pH study using a unique pH sensor into the pharynx (Restech^®^, Respiratory Technology Corp. San Diego, USA). This device is easy to use but, as for HEMII-pH, the analysis and the diagnosis criteria have to be standardized [2].

GI endoscopy has a limited role in the management of LPR. Primary care physicians may prescribe GI endoscopy in patients with heartburn, chest pain or GI symptoms but have to keep in mind that a normal GI endoscopy does not exclude the LPR diagnosis. As suggested in Table 2, elderly patients may have esophagitis without complaints; then, GI endoscopy may be useful for >50 years old patients with chronic symptoms.

The detection of pepsin in saliva may be possible in the primary care physician office through the peptest^®^ device (Peptest^TM^ kit; RD Biomed Ltd., Hull, United Kingdom). Patients have to collect two or three saliva samples and the physician performs the measurement of pepsin saliva concentration respecting a standardized procedure lasting 15 to 20 min. The physician used the Cube Reader^®^ that detects pepsin down to 16 ng/mL. As recommended [65], the test was considered as positive when the pepsin level reached 36 ng/mL. The pepsin saliva detection is easy to use but is still not validated and cannot be considered as a gold standard approach. Meta-analyses suggested that sensitivity and specificity of the peptest would be 64% and 68%, respectively [66,67]. There would have many grey areas limiting the establishment of clear indications for the peptest. First, the saliva pepsin concentration would be not correlated with the HEMII-pH findings [68]. Second, the diet of patients could have a significant impact on the pepsin saliva concentration [69]. Third, there is no consensus about the best time for saliva collection [67]. Some authors supported that pepsin has to be measured upon waking (morning) [69,70], but that has to be confirmed in future studies.

## 8. Treatment

### Cost-Effective Empirical Approach

Over the past few decades, the empirical therapeutic trial based on proton pump inhibitors (PPIs) was proposed as the main cost-effective approach to treat and support the LPR diagnosis [71,72,73,74,75,76,77,78,79,80,81,82,83,84]. Nowadays, this approach is increasingly challenged for many reasons [64,75]. First, PPIs are suspected to have short- and long-term side effects (Table 3) that support the PPI prescription only in patients with an identified acid reflux disease and for the shorter duration [64].

Second, the response to PPI does not guide the treating physician in how to proceed with non-responders, while it is possible that those with persistent cough, globus sensation, throat clearing, and/or other presumed LPR symptoms may actually not have LPR if they do not respond to empiric treatment. It is possible that refractory or alkaline LPR may be present; this would be identified by HEMII-pH but cannot be excluded on the basis of empiric treatment [2,64]. Alkaline and weakly acid LPR are more prevalent than previously presumed because they concern more than 50% of patients [76,77] and, therefore, require alginate therapy to control the alkaline component of reflux. Note than alginates are also interesting for GERD and acid LPR. Third, the PPI effects on LPR disease are still controversial since meta-analysis of placebo-RCTs did not find superiority of PPIs over placebo [12,78]. All of the arguments explain why the use of empirical PPI treatment is still controversial. In practice, an empirical treatment has to include diet, PPIs and alginate medication to ensure an efficacy on all types of LPR [64]. An adequate anti-reflux treatment may be helpful for the reflux symptoms but also for other conditions of the patients, such as sleep disorder [79], overweight [2] or dental disorders [51]. The primary care physician usually knows the lifestyle and the behavior of these patients. In that way, the physician could have a critical role in strengthening the relevance of diet in both the suspected and confirmed LPR disease. Because LPR is often due to diet habits [80,81] and stress [82], the primary care physician has a key role in alerting the patient about these favoring factors and preventing recurrence or chronicity of the disease. Some scores assessing the refluxogenic potential of diet were developed [83,84] and, through a mobile phone app, could be useful for patients in the choice of their favorite foods. Some foods and beverages are associated with a high risk of reflux while others are protective regarding LPR (Table 4 and Table 5). The awareness of patients regarding the importance of diet is crucial in the short to long-term management of LPR and the role of the primary care physician is crucial. Similar findings have to be considered for the management of stress and anxiety, which both lead to autonomic nerve dysfunction and transient esophageal sphincter relaxation [35,36,82]. A practical algorithm of management of LPR by primary care physician is proposed in Figure 2. In summary, to be cost-effective, a primary care physician may propose an empirical treatment based on diet and stress management for patients with mild LPR and no red flags. There are no consensual definitions of mild, moderate and severe LPR, but in this algorithm, we may define LPR as mild if the patient reports mild symptoms and low impact on quality of life. These patients may easily accept treatment with only diet and stress management. If the patient reports that the symptoms are troublesome, having a significant impact on quality of life, the LPR may be considered as moderate or severe and, therefore, the empirical treatment has to include PPIs and alginate for 2 to 3 months. All patients will not similarly respond to diet and behavioral changes [80,83,84]. Many patients with typical symptoms of GERD will require medication. Naturally, the algorithm has to be evaluated through clinical studies conducted in primary care medicine and could be improved in the next few years regarding new findings in the literature.

## 9. Conclusions

To date, it seems possible that a high number of GPs are still unaware of the entity of laryngopharyngeal reflux [86]. However, many LPR patients may be efficiently managed by primary care physicians if they consider using clinical tools describing symptoms and signs associated with LPR, excluding some confounding conditions and red flags and the use of an appropriate empiric treatment. In this study, we propose a practical algorithm to manage LPR in primary care medicine. The reliability of this algorithm has to be evaluated in future studies as well as the use of a peptest as a diagnostic method in the primary care practitioner office.

## Figures and Tables

**Figure 1 jcm-09-03618-f001:**
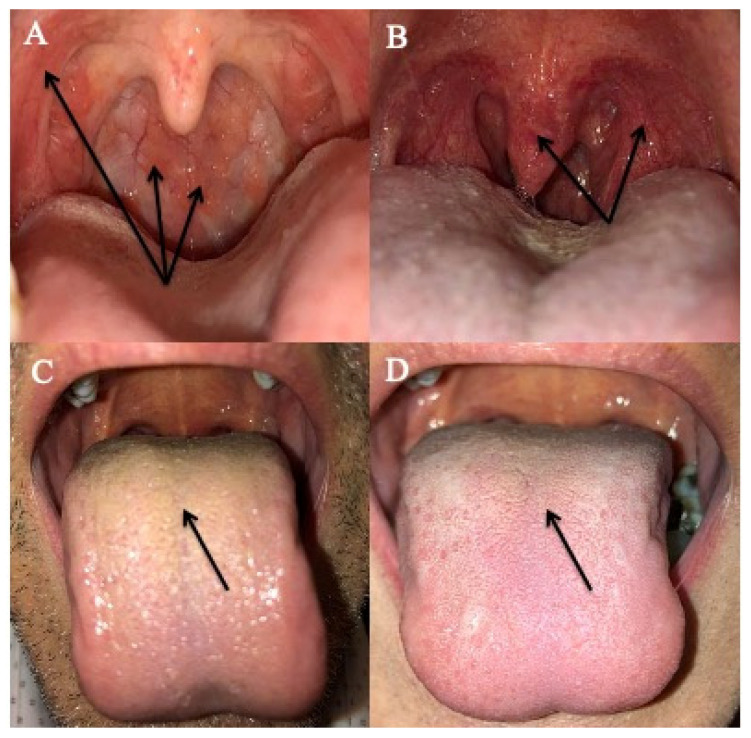
Oral and Oropharyngeal Findings Associated with LPR. Pharyngeal erythema (**A**), anterior pillar erythema (**A**,**B**) and uvula erythema (**B**) are signs easily identified in primary care practice (1,2) accounting for 89.5, 91.0 and 54% of cases. Coated tongue (**C**) is found in 49.4% of patients and may significantly improve through treatment (**D**). However, the primary care physician had to keep in mind that some patients have a significant improvement of symptoms but these signs may persist over time [55].

**Figure 2 jcm-09-03618-f002:**
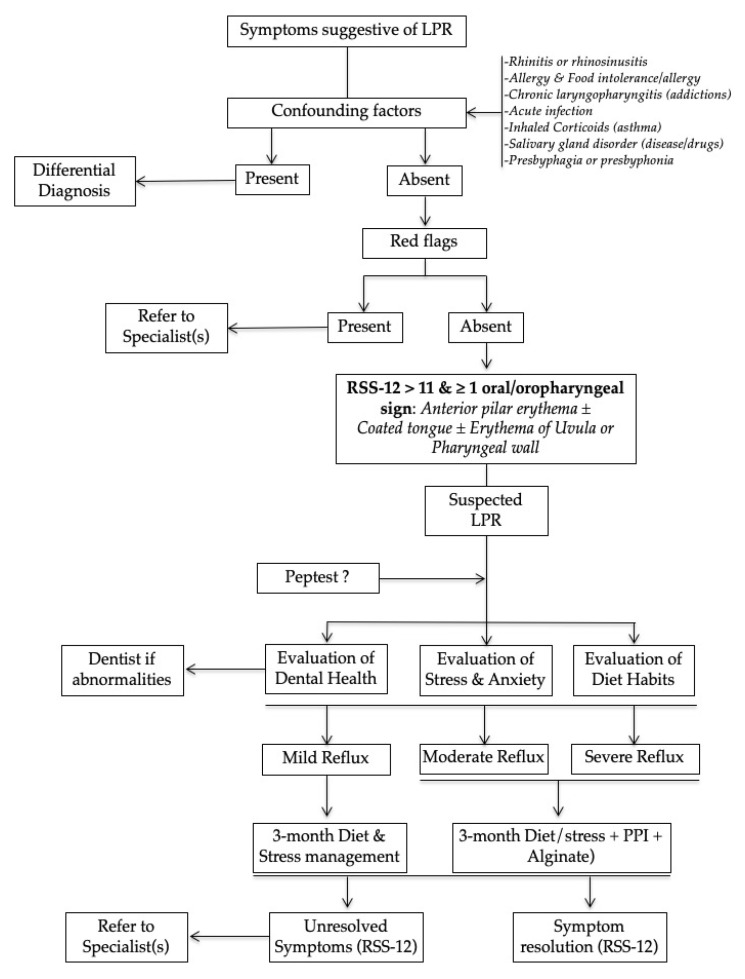
Practical algorithm of Management of Laryngopharyngeal Reflux in Primary care Medicine. Abbreviations: LPR = laryngopharyngeal reflux; PPI = proton pump inhibitors; RSS-12 = reflux symptom score-12.

**Table 1 jcm-09-03618-t001:** Reflux Symptom Score-12.

**Reflux Symptom Score-12**					
Within the last month, I suffered from one/several followed symptoms					
Severity: 0 = problem is not severe, 5 = problem very troublesome when it occurs					
Frequency: 0 = I don′t have this complaint over the past month, 1;2;3;4 = I had 1-2;2-3;3-4;4-5 weekly over the past month; 5 = complaint occurs daily					
	**Disorder Frequency**	**Disorder Severity**		**Quality of Life Impact**	
**Ear Nose and Throat Disorders**			Total score		Total score
1. Hoarseness or a voice problem	0 - 1 - 2 - 3 - 4 - 5	0 - 1 - 2 - 3 - 4 - 5	…………	0 - 1 - 2 - 3 - 4 - 5	…………
2. Throat pain or pain during swallowing time	0 - 1 - 2 - 3 - 4 - 5	0 - 1 - 2 - 3 - 4 - 5	…………	0 - 1 - 2 - 3 - 4 - 5	…………
3. Difficulty swallowing (pills, liquids or solid foods)	0 - 1 - 2 - 3 - 4 - 5	0 - 1 - 2 - 3 - 4 - 5	…………	0 - 1 - 2 - 3 - 4 - 5	…………
4. Throat clearing (not cough)	0 - 1 - 2 - 3 - 4 - 5	0 - 1 - 2 - 3 - 4 - 5	…………	0 - 1 - 2 - 3 - 4 - 5	…………
5. Sensation of something being stuck in the throat	0 - 1 - 2 - 3 - 4 - 5	0 - 1 - 2 - 3 - 4 - 5	…………	0 - 1 - 2 - 3 - 4 - 5	…………
6. Excess mucous in the throat and/or postnasal drip sensation	0 - 1 - 2 - 3 - 4 - 5	0 - 1 - 2 - 3 - 4 - 5	…………	0 - 1 - 2 - 3 - 4 - 5	…………
7. Bad breath	0 - 1 - 2 - 3 - 4 - 5	0 - 1 - 2 - 3 - 4 - 5	…………	0 - 1 - 2 - 3 - 4 - 5	…………
8. Heartburn, stomach acid coming up, regurgitations, burping or nausea	0 - 1 - 2 - 3 - 4 - 5	0 - 1 - 2 - 3 - 4 - 5	…………	0 - 1 - 2 - 3 - 4 - 5	…………
9. Abdominal pain or diarrhea	0 - 1 - 2 - 3 - 4 - 5	0 - 1 - 2 - 3 - 4 - 5	…………	0 - 1 - 2 - 3 - 4 - 5	…………
10. Indigestion, abdominal distension and/or flatus	0 - 1 - 2 - 3 - 4 - 5	0 - 1 - 2 - 3 - 4 - 5	…………	0 - 1 - 2 - 3 - 4 - 5	…………
11. Coughing (not just throat clearing)	0 - 1 - 2 - 3 - 4 - 5	0 - 1 - 2 - 3 - 4 - 5	…………	0 - 1 - 2 - 3 - 4 - 5	…………
12. Breathing difficulties, breathlessness or wheezing	0 - 1 - 2 - 3 - 4 - 5	0 - 1 - 2 - 3 - 4 - 5	…………	0 - 1 - 2 - 3 - 4 - 5	…………
		**RSS total score:**……… **Quality of Life score:**…………..			

Severity item (5-point) is multiplied by frequency (5-point) to obtain symptom score (0–25). The sum is calculated to obtain RSS-12 final score (0–300). A RSS-12>11 is suggestive of Laryngopharyngeal Reflux (LPR) and exhibits high sensitivity (94.5%) and specificity (86.2%) [50].

**Table 2 jcm-09-03618-t002:** Red flags requiring Specialist Consultation.

Red Flags that Support to Refer Patient to	
Otolaryngologist	Gastroenterologist
1. Onset of symptoms in alcohol drinkers/smokers.	1. Symptoms including severe heartburn and
2. symptoms in patients	chest pain.
with a history of head neck malignancy.	2. Symptoms including severe dysphagia,
3. Symptoms and neck nodes lasting for >3 weeks.	hypersalivation, or vomiting.
4. Weight loss without diet and lifestyle habit changes.	3. History of untreated Barrett metaplasia.
5. Aspirations and lung infections.	4. Chronic symptoms in patient >50 years old.
6. Voice professionals with severe dysphonia or	5. Unvoluntary weight loss >5% of weight.
patients with dysphonia lasting for >3 weeks.	6. Unexplained associated Iron deficiency.
7. Hemoptysis or dyspnea.	7. Gastrointestinal bleeding.
	8. Associated neck lymphadenopathy.
	9. Family history of upper digestive cancer.
	10. Non-response to empirical treatment.

**Table 3 jcm-09-03618-t003:** Long-term Side Effects of Proton pump inhibitors.

Systems	Presumed Side Effects of PPI	Status
Stomach	Increased risk of gastric neoplasia	Highly suspected
	Increased risk of Vitamin B12 deficiency	Suspected
	Increased risk of Calcium deficiency	Suspected
	Increased risk of Iron deficiency	Suspected
	Increased risk of Magnesium deficiency	Suspected
Digestive	Increased risk of bacterial, parasitic, and	Suspected
	fungal infections	
Liver	Increased risk of Cancer	Suspected
	Increased risk of Bacterial overgrowth	Suspected
Kidney	Increased risk of Acute Interstitial Nephritis	Highly suspected
	Increased risk of Chronic kidney disease	Suspected
Bone	Increased risk of Osteoporosis and fracture	Suspected
Brain	Increased risk of Dementia	Suspected
Chest	Increased risk of pneumonia *	Suspected
Cardiovascular	Increased risk of cardiovascular events **	Suspected
	Increased risk of electrolyte imbalances	Suspected

The association between proton pump inhibitors and many disorders is suspected or highly suspected [70]. * The association between PPI use and pneumonia risk was particularly found in elderly patients, patients admitted in intensive care units with dementia, with a history of acute stroke, type 2 diabetes or cirrhosis and those with chronic GERD. ** There will be an interaction between clopidogrel and PPIs; underlying the increased risk of cardiovascular events in patients who take clopidogrel and PPIs. Abbreviations: GERD = gastroesophageal reflux disease; PPI = proton pump inhibitor. Abbreviations: PPISs = proton pump inhibitors.

**Table 4 jcm-09-03618-t004:** The Refluxogenic Diet Score of foods and their Refluxogenic Potential.

Very Low Reflux. Foods	REDS	Cat.	Low Reflux. Foods	REDS	Cat.	Moderate Reflux. Foods	REDS	Cat.	High Reflux. Foods	REDS	Cat.	Very High Reflux. Foods	REDS	Cat.
Artichoke	0.086	1	Aubergine	0.166	2	Apricot	0.391	3	Apple	0.534	4	Avocado	5.610	5
Asparagus *	0.072	1	Banana	0.227	2	Blueberry	0.472	3	Blackberries	0.640	4	Bacon	25.40	5
Baked spinach	0.025	1	Carrots	0.132	2	Boiled egg	0.348	3	Brie, Blue, bread cheeses	1.001	4	Butter	-	5
Beetroot	0.082	1	Cherry	0.243	2	Camembert	0.495	3	Cake	1.850	4	Candy or sweets	5.216	5
Broccoli	0.077	1	Chicken fillet	0.148	2	Cereals (corn flacks)	0.470	3	Cauliflower	0.596	4	Chocolate (dark)	4.171	5
Brussels sprout	0.030	1	Chili	0.171	2	Courgettes	0.289	3	Cheddar	1.068	4	Chocolate (Milk)	3.787	5
Celery	0.101	1	Corn	0.244	2	Cucumber	0.274	3	Chocolate cookies	1.920	4	Chocolate (white)	4.543	5
Cooked mushrooms	0.103	1	Fat chicken	0.236	2	Dried plum	0.252	3	Cookies	1.695	4	Chocolate croissant	2.911	5
Crabs	0.088	1	Fennel	0.131	2	Duck (without skin and fat)	0.350	3	Cracker	0.952	4	Chocolate eclairs	2.079	5
Egg white	0.006	1	Ketchup **	0.166	2	Fat fish	0.368	3	Egg yolk	1.334	4	Croissant	2.860	5
Endive	0.014	1	Kidneys	0.192	2	Fig	0.267	3	Feta	1.501	4	Curry	2.985	5
Fresh and thin fish	0.058	1	Lamb	0.232	2	Fish oil (sardines, cods)	-	3	Fontina	0.946	4	French fries and frying	2.836	5
Garlic	0.035	1	Lamb chops or shoulder	0.201	2	Fish oil (herrings)	-	3	Goat cheese	1.061	4	Ice cream	3.364	5
Green beans	0.054	1	Leek	0.139	2	Fish sauce	0.428	3	Gouda	1.193	4	Macadamia nut	7.074	5
Green peas	0.095	1	Melon	0.189	2	Ginger	0.362	3	Ground meat	0.704	4	Mayonnaise	56.80	5
Green salad *	0.074	1	Oat	0.243	2	Grapefruit	0.392	3	Gruyere	0.992	4	Meat sauce (Bearnaise)	45.04	5
Honey	0.000	1	Onion *	0.129	2	Guava	0.376	3	Hard cheese, full-fat cheese	1.093	4	Meat sauce (Pepper)	3.839	5
Horse	0.076	1	Parsley	0.139	2	Lamb cutlets	0.462	3	Kiwi	0.540	4	Meat sauce (Roquefort)	3.060	5
Lentil	0.064	1	Pepper	0.186	2	Mandarin	0.478	3	Lychee	0.512	4	Milk (coco)	6.521	5
Low-fat cheese	0.003	1	Pork tenderloin	0.208	2	Milk (goat, semi-skimmed)	0.272	3	Mango	0.536	4	Nut, cashew, hazelnut	3.585	5
Milk (Skimmed)	0.030	1	Rib steak	0.153	2	Milk (soja)	0.298	3	Meat sauce (Mushroom)	1.116	4	Olive (black)	7.478	5
Mollusk	0.060	1	Ribs	0.246	2	Milk (Semi-skimmed)	0.363	3	Milk (whole)	0.690	4	Oliver (green)	12.92	5
Pork roast	0.110	1	Rice (Brown)	0.188	2	Mint	0.302	3	Mozzarella	1.025	4	Pasta sauce (carbonara)	2.071	5
Pumpkin	0.085	1	Rindless, fatless,	0.131	2	Nectarine	0.292	3	Munster	1.223	4	Pasta sauce (pesto)	8.331	5
Red cabbage	0.046	1	Cooked ham	0.131	2	Olive oil	-	3	Mustard	1.839	4	Pesto	8.331	5
Rice (Red)	0.121	1	Rye bread	0.166	2	Orange	0.381	3	Noodles	0.565	4	Potato chips	2.830	5
Rice (White)	0.089	1	Shallot *	0.201	2	Peach	0.361	3	Orange jam	0.623	4	Sauerkraut	5.696	5
Roast veal	0.090	1	Steak, fillet, striploin	0.208	2	Pear	0.364	3	Parmesan	0.836	4	Spicy ##	0.000	5
Shrimps or lobster	0.033	1	Tofu	0.248	2	Pickle	0.270	3	Pasta sauce (Bolognese)	1.134	4			
Spaghettis (cooked)	0.060	1	Turnip	0.186	2	Plum	0.471	3	Pâté	1.612	4			
Sweet potato	0.073	1	Veal chop	0.181	2	Pork chops and shoulder	0.316	3	Peanut	1.618	4			
Tuna (low-fat)	0.043	1	Watermelon	0.175	2	Potato	0.357	3	Pomegranate	0.725	4			
Turkey fillet	0.026	1	White bread	0.187	1	Raspberry	0.307	3	Raisin	0.758	4			
Veal cutlet	0.059	1	Whole ham	0.236	2	Rhubarb	0.362	3	Raspberry jam	0.566	4			
Wheat	0.079	1				Salmon	0.375	3	Redcurrant	0.922	4			
						Sardines	0.290	3	Ricotta	1.030	4			
						Strawberry	0.340	3	Roquefort	1.288	4			
						Sugar #	0.000	3	Salami	1.177	4			
						Tomato (raw)	0.297	3	Sausages	0.722	4			
						Tripes	0.255	3	Sorbet	1.942	4			
						Whole meal/brown bread	0.264	3	Strawberry jam	0.618	4			
									Tomato sauce	1.538	4			
									Vinaigrette	-	4			
									Yoghurt (fat)	0.674	4			

Categories 1 and 2 correspond to low refluxogenic foods while categories 4 or 5 include foods with a high or very high refluxogenic potential [84,85]. Some foods may be upgraded or downgraded regarding to characteristics. * Raw vegetables are less digestible and may be associated with low gastric emptying time: in case of raw consumption, the food has to be upgraded for 1 category. Not for green salad, the addition of vinegar or vinaigrette upgrades the category. ** In case of addition of spicy (for example, Spicy Ketchup), these foods have to be upgraded. # For sugar, only the pH and the glycemic index have been considered regarding the lack of fat. ## Because spicy has no lipid and no pH, the authors based the classification of this food on the literature. If the patients only eat industrial foods (ready-made food), the foods may be upgraded regarding the acidifying potential of industrial conservative. Abbreviations: REDS = refluxogenic diet score.

**Table 5 jcm-09-03618-t005:** The Refluxogenic Diet Score of beverages and their related categories.

Juice, Water and Alcohol	pH	GI > 40	Cat.	UCat.
Alcohol (strong and licor) *°	4	+	3	5
Aloe vera	6.1	0	2	2
Apple juice	3.65	+	4	5
Beer #(°)	4	+	3	5
Cacao (hot chocolate)	6.3	+	2	3
Chamomile	6.5	0	2	2
Chicory	5.95	0	3	3
Coffee **	5	0	3	4
Grapefruit juice	3.05	+	4	5
Lemon juice	2.3	+	4	5
Multifruit juice	3.8	+	4	5
Orange juice	3.5	+	4	5
Soda (sugar free) #	2.5	0	4	5
Soda (with sugar) #	2.5	+	4	5
Syrup (Mint, lemon, grenadine)	2.15	+	4	5
Tea **	5	0	3	4
Tea (blackberry) **	2.5	0	4	5
Tea (black) **	5.3	0	3	4
Tea (green) **	7	0	2	3
Tea (lemon) **	2.9	0	4	5
Tomato juice	4.35	0	3	3
Water (sparkling) #	7	0	2	3
Water (still)	7	0	2	2
Water (alkaline)	8	0	1	1
Wine (red) °	4	0	4	5
Wine (rose) °	4	0	4	5
Wine (white) °	4	0	4	5

The classification of beverages depends on pH, * glycemic index (GI; high sugar-related osmolarity), # sparkling (upgrade), ° the alcohol degree (>3% = upgrade) and the ** presence or lack of caffeine or theine (** upgrade or downgrade). Abbreviations: GI = glycemic index; cat. = category at baseline; ucat. = upgraded category. For hot chocolate, the category is upgraded in case of additional sugar.

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
