# Peer review of "Laryngopharyngeal Reflux: A State-of-the-Art Algorithm Management for Primary Care Physicians"

_jcm, 2020, doi:10.3390/jcm9113618_

Round 1

Reviewer 1 Report

The article "Management of Laryngopharyngeal Reflux: A Practical Algorithm Management for Primary care Physicians" provides a thorough summary of the current knowledge and thought processes regarding diagnosis and treatment of LPR geared toward primary care physicians. The authors provide a thorough review of the literature in this regard throughout the paper, although it does refer to a lot of the author groups own publications. While I don't find this excessive (as it is a bit of a "how we do it" type of manuscript), this may be of concern to the editors.

The article covers just about every topic I would consider worthy of coverage for this topic and includes fairly extensive tables and figures that are quoted from the author's prior manuscripts (reflux dietary values, oropharyngeal findings in reflux). The authors provide a relatively succinct flow chart at the end that discusses care paradigms for these patients and provides guidance on whether empiric treatment is indicated.

Overall, my main complaint with this manuscript is with the overall English grammar and style throughout from this European author group. The writing is fairly poor throughout and this makes the article difficult to read at times. These are all fairly easily corrected grammatical issues but they are extensive and throughout the manuscript.

My other main concern with this paper is that I feel it perhaps too detailed as a summary intended for primary care physicians. The author group is certainly choosing to be as thorough as possible, but it is perhaps not necessary to cover all of this in one review article. They may be better suited by focusing on which patients to choose to treat for presumed LPR (or to refer), and what appropriate treatment options should be. This message is somewhat lost in the length and wordiness of this manuscript.

Author Response

Reviewer 1:

The article "Management of Laryngopharyngeal Reflux: A Practical Algorithm Management for Primary care Physicians" provides a thorough summary of the current knowledge and thought processes regarding diagnosis and treatment of LPR geared toward primary care physicians. The authors provide a thorough review of the literature in this regard throughout the paper, although it does refer to a lot of the author groups own publications. While I don't find this excessive (as it is a bit of a "how we do it" type of manuscript), this may be of concern to the editors. 
Our international study group of reflux wrote 60 articles over the previous 6 years. Many of these studies reported significant findings that may be helpful to write this state of the art review and provided tip management for primary care or non-ENT physicians.

The article covers just about every topic I would consider worthy of coverage for this topic and includes fairly extensive tables and figures that are quoted from the author's prior manuscripts (reflux dietary values, oropharyngeal findings in reflux). The authors provide a relatively succinct flow chart at the end that discusses care paradigms for these patients and provides guidance on whether empiric treatment is indicated. 
Thank you.

Overall, my main complaint with this manuscript is with the overall English grammar and style throughout from this European author group. The writing is fairly poor throughout and this makes the article difficult to read at times. These are all fairly easily corrected grammatical issues but they are extensive and throughout the manuscript. 
A native speaker EN corrected the manuscript to improve grammar and spelling.

My other main concern with this paper is that I feel it perhaps too detailed as a summary intended for primary care physicians. The author group is certainly choosing to be as thorough as possible, but it is perhaps not necessary to cover all of this in one review article. They may be better suited by focusing on which patients to choose to treat for presumed LPR (or to refer), and what appropriate treatment options should be. This message is somewhat lost in the length and wordiness of this manuscript.

We significantly reduced the size of the paper according to the comment of the reviewer.

We especially reduce the size of the following paragraphs:

-Prevalence & Incidence.

-Pathophysiology.

-Many grey area.

-Additional examinations.

We remove Table 1.

Reviewer 2 Report

This is a comprehensive and thorough review on laryngopharyngeal reflux, addressing the issues more relevant to general practitioners. Scientific concepts are up todate and a meaningful management algorithm  is presented focusing on the need to adopt dietary restrictions along with medication.

The topic of laryngopharyngeal reflux has been heavily researched and many research groups have published original research results that could be cited. 

ref 50 needs to be corrected 

Author Response

This is a comprehensive and thorough review on laryngopharyngeal reflux, addressing the issues more relevant to general practitioners. Scientific concepts are up todate and a meaningful management algorithm  is presented focusing on the need to adopt dietary restrictions along with medication.

The topic of laryngopharyngeal reflux has been heavily researched and many research groups have published original research results that could be cited. 

ref 50 needs to be corrected 

We corrected reference 50: Murray RC, Chennupati SK. Chronic streptococcal and non-streptococcal pharyngitis. Infect Disord Drug Targets. 2012; 12(4):281-5. doi: 10.2174/187152612801319311..

Reviewer 3 Report

The authors try to propose a simple and practical algorithm that, in their opinion, can be useful to manage patients with suspected LPR in daily clinical practice.

They correctly discuss on the difficult diagnosis of LPR, due to the fact it is frequently based on symptoms, which are subjective and shared by other diseases of the upper respiratory tract. Unfortunately, also objective findings we can obtain with laryngoscopy are highly aspecific and can be found in normal subjects (Hicks DM et al, 2002).

The algorithm they propose for general practitioners has many weak points, which are similar to those already described. For instance, the detection of oral/pharyngeal signs of inflammation remains subjective and a study comparing them in patients with suspected reflux and healthy controls should be done in order to assess their accuracy, particularly in terms of false positive results. The use of pep-test, although intriguing, is not considered at present as reliable for the diagnosis of GERD. The differentiation among mild, moderate and severe reflux is very difficult from a practical point of view and finally, the therapeutic value of diet has been shown to be low even in patients with typical reflux symptoms.

Therefore, I suggest the authors to be very cautious in proposing their algorithm and to attenuate as much as possible the value of its diagnostic reliability in condideration of the lack of rigorous studies in this field.

Author Response

The authors try to propose a simple and practical algorithm that, in their opinion, can be useful to manage patients with suspected LPR in daily clinical practice. They correctly discuss on the difficult diagnosis of LPR, due to the fact it is frequently based on symptoms, which are subjective and shared by other diseases of the upper respiratory tract. Unfortunately, also objective findings we can obtain with laryngoscopy are highly aspecific and can be found in normal subjects (Hicks DM et al, 2002).

We agree. We added this information and the related reference to the finding paragraph: p.7, findings, line 5: « The interrater reliability of RFS is low, especially regarding the nonspecificity of signs [57], which limits the reproducibility between otolaryngologists [58]. As reported by Hicks et al., some LPR-associated findings may be found in normal individuals [57]”

The algorithm they propose for general practitioners has many weak points, which are similar to those already described. For instance, the detection of oral/pharyngeal signs of inflammation remains subjective and a study comparing them in patients with suspected reflux and healthy controls should be done in order to assess their accuracy, particularly in terms of false positive results.

This study was already performed and published: in the validation paper of Reflux Sign Assessment, authors found significant higher prevalence of these signs in LPR patients regarding HEMII-pH compared with healthy individuals (Lechien JR, Validity and Reliability of the Reflux Sign Assessment. Ann Otol Rhinol Laryngol, 2019). However, we agree that additional studies could be useful to confirm these data.
Therefore, we added this information: p.7, findings, line 12: LPR patients have a significant higher prevalence of anterior pilar erythema, coated tongue, uvula and oropharyngeal posterior wall erythema compared with healthy individuals [60]. These signs may be easily seen by the primary care physician and may be useful for both the diagnosis and the posttreatment follow-up (Figure 1) [61]. However, primary care physician has to keep in mind that the assessment of findings is still subjective supporting that the pre- to posttreatment evaluation needs to be performed by the same physician.»

The use of pep-test, although intriguing, is not considered at present as reliable for the diagnosis of GERD.

We temperate the peptest paragraph.p.10, paragraph 3: “The detection of pepsin in saliva may be possible in the primary care physician office through the peptest® device (PeptestTM kit; RD Biomed Ltd., Hull, United Kingdom). Patient has to collect 2 or 3 saliva samples and the physician performs the measurement of pepsin saliva concentration respecting a standardized procedure lasting 15 to 20 minutes. The physician used the Cube Reader® that detects pepsin down to 16 ng/mL. As recommended [72], the test was considered as positive when the pepsin level reached 36 ng/mL. The pepsin saliva detection is easy to use but is still not validated and cannot be considered as a gold standard approach. Meta-analyses suggested that sensitivity and specificity of peptest would be 64% and 68%, respectively [73,74]. There would have many grey areas limiting the establishment of clear indications for peptest. First, the saliva pepsin concentration would be not correlated with the HEMII-pH findings [75]. Second, the diet of patients could have a significant impact on the pepsin saliva concentration [76]. Third, there is no consensus about the best time of saliva collection [74]. Some authors supported that pepsin has to be measured upon waking (morning) [76,77] but that has to be confirmed in future studies.”

The differentiation among mild, moderate and severe reflux is very difficult from a practical point of view and finally, the therapeutic value of diet has been shown to be low even in patients with typical reflux symptoms. Therefore, I suggest the authors to be very cautious in proposing their algorithm and to attenuate as much as possible the value of its diagnostic reliability in condideration of the lack of rigorous studies in this field.

We discuss about these points in the treatment paragraph, p.12, last paragraph: “There are no consensual definitions of mild, moderate and severe LPR but, in this algorithm, we may define LPR as mild if patient reports mild symptoms and low impact on quality of life. These patients may easily accept to be treated with only diet and stress management. If the patient reports that the symptoms are troublesome, having a significant impact on quality of life, the LPR may be considered as moderate or severe and, therefore, the empirical treatment has to include PPIs and alginate for 2 to 3 months. All patients will not similarly respond to diet and behavioral changes [87,90,91]. Many patients with typical symptoms of GERD will require medication. Naturally, the algorithm has to be evaluated through clinical studies conducted in primary care medicine and could be improved in the next few years regarding new findings in the literature.”

Round 2

Reviewer 3 Report

The authors have answered satisfactorily to my comments and have reduced the emphasis on the value of their diagnostic and therapeutic algorithm.